# In situ infrared nanospectroscopy of the local processes at the Li/polymer electrolyte interface

Xin He[1,3,4], Jonathan M. Larson [1,4,5✉], Hans A. Bechtel [2,5✉] & Robert Kostecki [1,5✉]

Solid-state batteries possess the potential to significantly impact energy storage industries by enabling diverse benefits, such as increased safety and energy density. However, challenges persist with physicochemical properties and processes at electrode/electrolyte interfaces. Thus, there is great need to characterize such interfaces in situ, and unveil scientific understanding that catalyzes engineering solutions. To address this, we conduct multiscale in situ microscopies (optical, atomic force, and infrared near-field) and Fourier transform infrared spectroscopies (near-field nanospectroscopy and attenuated total reflection) of intact and electrochemically operational graphene/solid polymer electrolyte interfaces. We find nanoscale structural and chemical heterogeneities intrinsic to the solid polymer electrolyte initiate a cascade of additional interfacial nanoscale heterogeneities during Li plating and stripping; including Li-ion conductivity, electrolyte decomposition, and interphase formation. Moreover, our methodology to nondestructively characterize buried interfaces and interphases in their native environment with nanoscale resolution is readily adaptable to a number of other electrochemical systems and battery chemistries.

[1] Energy Storage and Distributed Resources Division, Lawrence Berkeley National Laboratory, Berkeley, CA 94720, USA. [2] Advanced Light Source, Lawrence Berkeley National Laboratory, Berkeley, CA 94720, USA. [3] Present address: School of Chemical Engineering, Sichuan University, 610017 Chengdu, PR China. [4] These authors contributed equally: Xin He, Jonathan M. Larson. [5] These authors jointly supervised this work: Jonathan M. Larson, Hans A. Bechtel, Robert Kostecki. ✉email: jmlarson@lbl.gov; HABechtel@lbl.gov; R_Kostecki@lbl.gov

Lithium (Li) solid-state batteries (SSBs) are a compelling electrical energy storage technology for future applications in portable electronics and transportation, particularly as continued advances demonstrate improvements in energy and power density, cycle-life, safety, and cost[1–3]. SSBs with a polymer electrolyte offer additional key benefits, including low self-discharge rate, different shapes according to device design, and compatibility with large-scale roll-to-roll manufacturing[4,5]. However, SSBs still must overcome performance limitations associated with electrode/electrolyte interfaces, such as high interfacial impedance[6], electrochemical instability[7], and inhomogeneous Li plating and stripping[6,8]. Including and beyond these specific challenges, it is hard to overstate the central role that electrode/electrolyte interfaces hold. In fact, battery performance is largely determined by thermodynamic, kinetic, and mechanical properties of such electrochemical interfaces[2,9,10]. Furthermore, in these electrochemical systems that operate far away from equilibrium, a thin passive film forms within the electrode/electrolyte interface. Situated at the surface of the electrode, and formed during battery assembly and/or operation, this so-called solid electrolyte interphase (SEI) layer is critical. It provides sufficient electronic resistivity and ionic conductivity to inhibit side reactions with the electrolyte while still allowing Li$^+$ transport. Additionally, the SEI's inhomogeneous structure and chemistry influence localized current density distribution and the resulting Li nano/micro-morphology evolution during charge–discharge processes.

However, our overall understanding of heterogeneous ionic interfaces and interphases is still very limited due to two main reasons. First, characterizing such interfaces and interphases in their native environment is extremely challenging, as they are buried between two dissimilar materials. Second, the interfaces and interphases have complex structure and chemistry[10], and can even evolve by chemical inter-diffusion, lattice strain, defects, and space charge effects which lead to a variety of chemical reactions across multiple spatial and temporal scales. Interface evolution is further propelled by the large amount of charge and mass transfer between electrodes in a battery over its lifetime (generally leading to degradation, performance loss, and eventual battery failure[11]).

The importance of interfaces and interphases, in combination with our limited understanding brought about by the challenges outlined above, continues to motivate the need for the development and application of effective in situ diagnostic approaches, especially those capable of characterizing interfacial structure, chemistry, and crystallinity over microscale regions with nanoscale spatial resolution[2,3,9,10], that is, at the level of the film's individual building blocks. A variety of techniques have been used to characterize the SEI, including electron[12] and cryogenic electron microscopy[13], optical techniques[14], X-ray absorption, and photoelectron spectroscopy[15,16], neutron depth profiling[17], and X-ray computed tomography[18]. However, to our knowledge no methodology has been demonstrated thus far that can characterize interface and SEI nondestructively, within their undisturbed native environment, and with nanoscale resolution. Here, we present an in situ methodology capable of just that.

In this work, we exploit the nanoscale spatial resolution, chemical selectivity, and surface sensitivity of near-field infrared nanospectroscopy to characterize graphene/Li/solid polymer electrolyte (SPE) interfaces. The single-layer graphene sheet operates as both an infrared transparent window and a current collector for Li plating and stripping (similar in function to Li anode-free batteries)[19,20]. Near-field infrared measurements reveal that intrinsic molecular, structural, and chemical heterogeneities in the SPE lead to the nonuniform Li plating and formation of a mosaic-like solid electrolyte interphase at the Li/SPE interface on a similar length scale. This study provides a unique insight into the mechanisms of early-stage interphase formation at electrochemically active buried interfaces, and an experimental diagnostic means to aid in the development of methods to control local nanoscale variations in electrolyte chemistry, structure, and ionic conductivity at the surface of the electrode.

## Results and discussion

**Experimental setup and characterization approach**. We utilize a model graphene-capped solid-state polymer battery cell in combination with attenuated total reflection Fourier transform infrared spectroscopy (ATR-FTIR) and near-field infrared nanospectroscopy (nano-FTIR). The near-field IR imaging and spectroscopy, which are based on scattering scanning near-field optical microscopy, are particularly attractive because they break the classic diffraction limit, allowing shallow nanoscale probing, spatial resolution at ca. 10 nm, and sensitivity to local molecular composition and orientation (details in methods). These near-field-based characterization tools have been recently harnessed to study a breadth of materials sciences topics relevant to biology[21], catalysis[22], plasmonics[23], information storage[24], solid-state physics[25], energy storage[26–28], and basic materials science of solid/liquid interfaces[29].

A three-dimensional schematic of a custom model cell for in situ nano-FTIR probing of graphene/Li/solid polymer electrolyte interfaces is shown in Fig. 1a. The device is comprised of single-layer graphene (G) atop of polyethylene oxide (PEO) and Li bis(trifluoromethanesulfonyl)imide (LiTFSI) solid electrolyte (EO:Li$^+$ 10:1 wt., see Methods for details) and a piece of Li foil, which operates as a counter and reference electrode. This configuration allows the graphene to serve as a conductive working electrode for Li plating, similarly to Li anode-free batteries, and at the same time act as an IR transparent window. Thus, chemical and structural evolution of the electrochemical interface can be characterized in situ at nanoscale subsurface depths[29,30] with the near-field IR probes applied from the outside of the G window, whereas the bulk SPE can similarly be sensed with ATR-FTIR at microscale subsurface depths[31].

Interfacial and bulk electrolyte measurements were carried out at four different stages of the charge-discharge process as depicted in Fig. 1b (also see Methods): (i) pristine G/SPE, (ii) after heating to 45 °C, (iii) after plating Li on graphene, and (iv) after stripping Li from graphene. The Li plating and stripping processes were performed galvanostatically ($i = +/-25\ \mu A\ cm^{-2}$) at 45 °C. Heating is employed to increase Li$^+$ conductivity of the polymer electrolyte as higher temperature enhances molecular movement and lowers the energy barrier for Li$^+$ transport[5]. At the end of each stage, once room temperature had been reached and device leads were disconnected, atomic force microscopy (AFM) images, broadband IR "white light" (WL) images, near-filed nano-FTIR spectra, and ATR-FTIR spectra were recorded. These complementary measurements allowed characterization of G/Li/polymer electrolyte interface/interphase structure, morphology, chemical composition, and crystallinity; as well as the chemistry and crystallinity of the bulk electrolyte. All experimental work was performed in Ar- or N$_2$-filled gloveboxes and environmental chambers to prevent exposure to air (see Methods and Supplementary Fig. 1).

**Characterization when pristine and after heating**. When a polymer surface contacts a smooth but elastic single-layer graphene sheet, adhesion between the two surfaces allows graphene to reproduce the morphology of the underlying polymer. The AFM and WL images of the pristine G/SPE interface at room temperature (Fig. 2a) reveal a fine structure of agglomerated high aspect ratio (80–500 nm) grains of the polymer electrolyte. Similar measurements performed after heating to 45 °C reveal

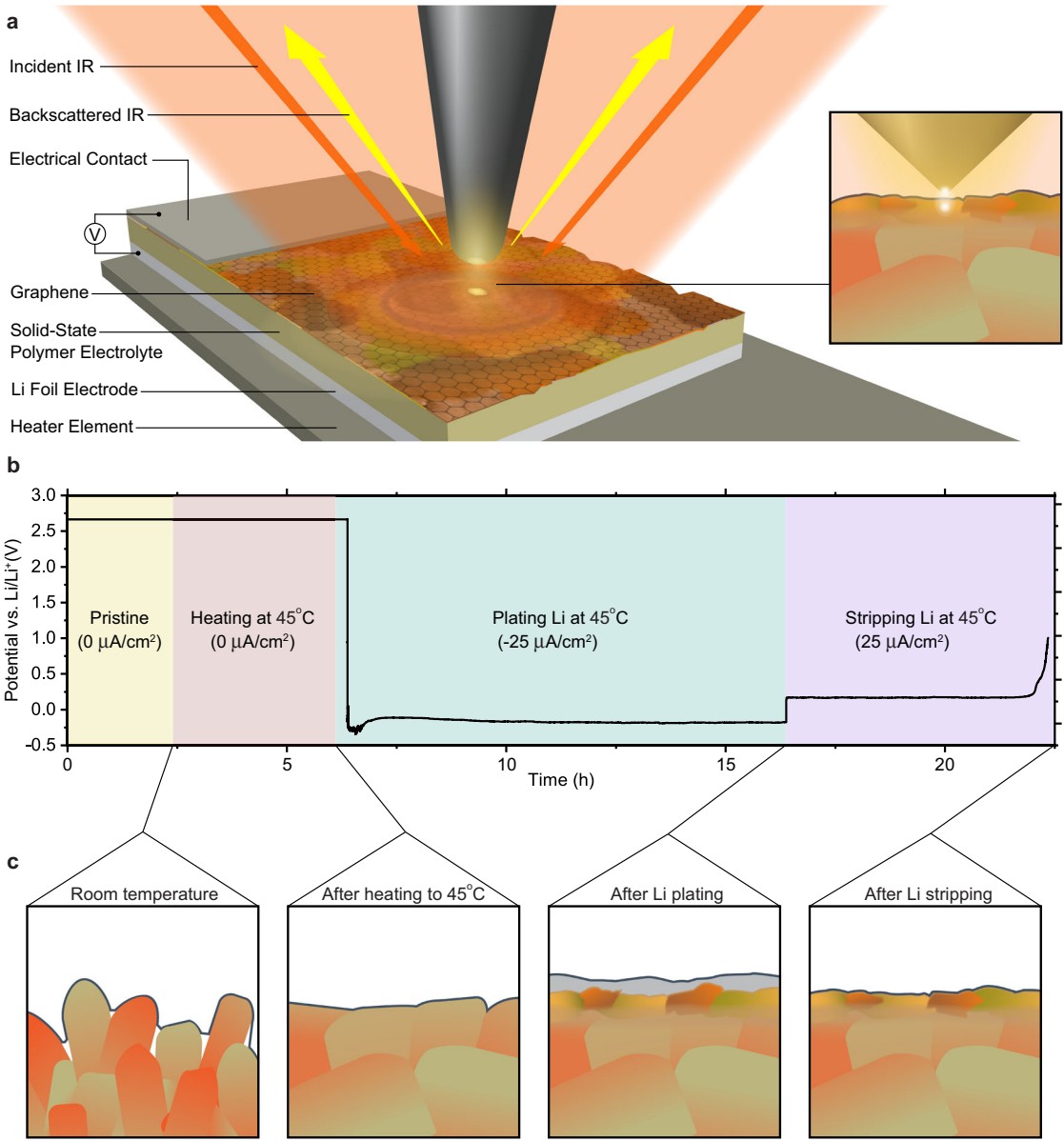

**Fig. 1 Experimental setup and methodology for in situ near-field infrared imaging and nanospectroscopy of the G/Li/polymer electrolyte interface.**
**a** 3-D schematic of the model battery cell atop heater element, including nano-FTIR probe (insert shows zoom in view of the probed area). **b** The electrode potential with respect to Li/Li$^+$ vs. time profile for the experiment. **c** Cross-sectional illustrations of the interfacial region at four stages of cell operation.

that the SPE grains coalesced, forming larger clusters (1–5 μm) (Fig. 2b). Furthermore, the SPE surface became smoother with root mean square roughness values dropping from 56 to 27.5 nm. This "smoothing" effect is associated with reduced viscosity and enhanced fluidity of the polymer, which becomes more mobile as temperatures approach the glass transition temperature.

Local nano-FTIR spectra of the G/SPE interface at room temperature and after heating to 45 °C (Fig. 2c, d) were recorded at locations indicated in Fig. 2a, b (dot's color matches the corresponding nano-FTIR spectra). All band assignments of TFSI$^-$ and PEO are summarized in Supplementary Figs. 2, 3 and Supplementary Table 1. Interestingly, the relative intensities of absorption peaks vary with location. These observed intensity variations in TFSI$^-$ and PEO IR absorption bands likely arise from a combination of local variations in PEO chain structure and orientation[32], relative TFSI$^-$ molecular conformations and/or orientations[33,34], and local salt (LiTFSI) concentration[35] (see Methods and Supplementary Fig. 4).

The extent of local chemical and molecular structure variation in the electrolyte is represented in Fig. 2e for the six selected vibrational modes as a percent of deviation from the mean peak intensity (details in Methods and, for comparison purposes, results from a control experiment conducted on SiO$_2$ are provided in Supplementary Fig. 6). The overlaid shadowed areas represent standard deviations above and below the means of the collective datasets. Heating of the cell significantly reduces the nanoscale chemical variability of the polymer electrolyte at the graphene surface (standard deviation decreases by 34%) but does not eliminate it altogether. These findings, along with the aforementioned variations in polymer chain orientations and salt-polymer molecular conformations, may imply that the SPE in the vicinity of the electrode is comprised of a patchwork of nanoscale regions of varying chemical compositions and molecular structures. In fact, Li$^+$ transport in PEO-based SPEs is believed to occur along the -C-O-C- units of the polymer chains[5] and the observed local structural molecular variations in PEO may influence and alter local Li$^+$ conductivity. Moreover, this effect could

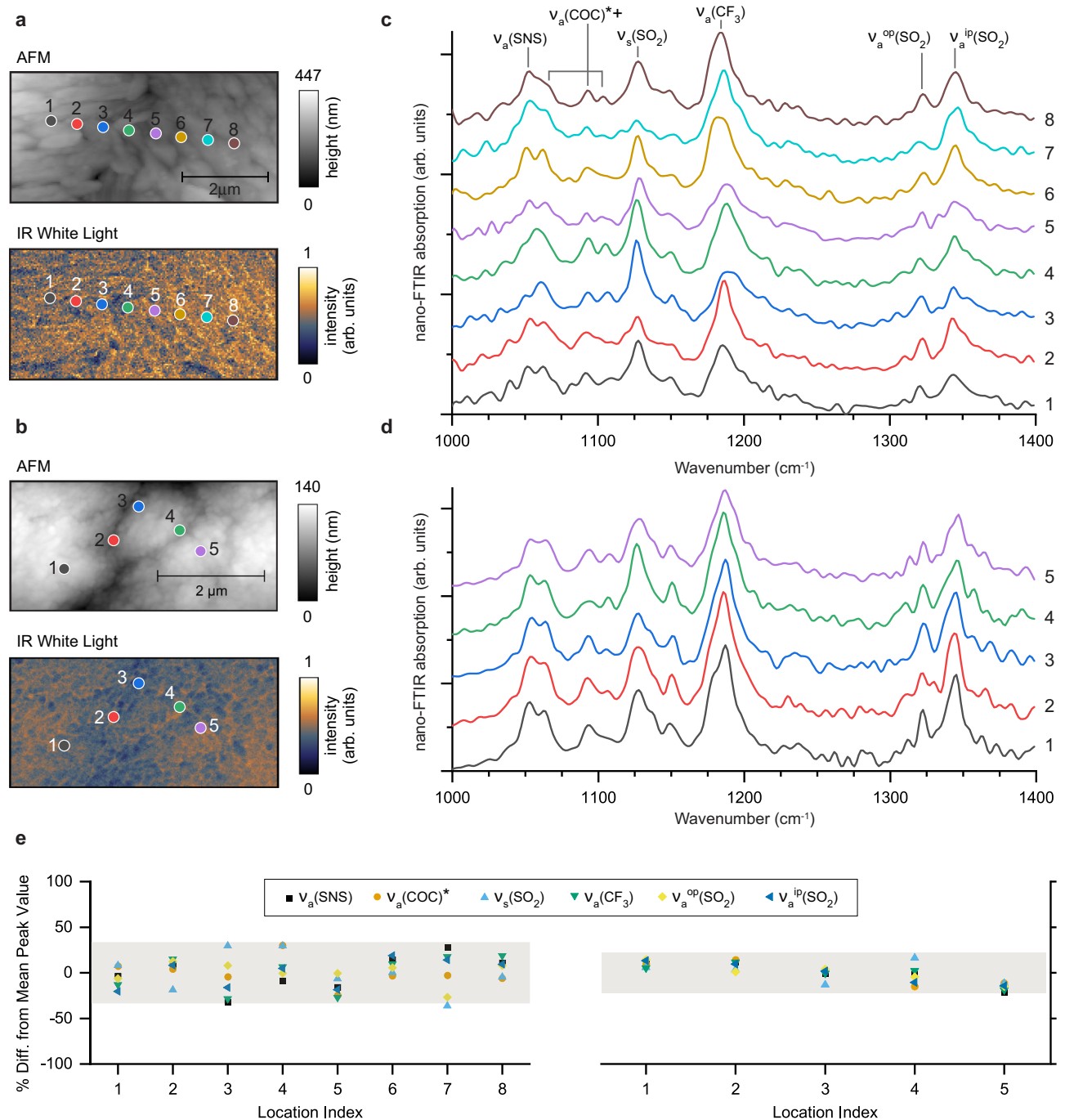

**Fig. 2 Characterization of the pristine G/SPE interface at room temperature and after heating to 45 °C.** Atomic force microscopy (AFM), infrared (IR) white light images, and local nano-FTIR spectra at room temperature (**a** and **c**) and after heating to 45 °C (**b** and **d**). **e** Scatterplots of six IR bands as a percentage of mean peak height at room temperature and 45 °C, plotted as a function of probing location. Overlaid shadowed areas indicate plus/minus two standard deviations for data sets.

be further enhanced by the observed local variations of LiTFSI salt concentration (see Methods and Supplementary Fig. 4) in that ionic conductivity in SPEs is connected to salt concentration[5,36]. Importantly, these nanoscale chemical inhomogeneities in the electrolyte can also slightly modify the local mechanism and rate of electrolyte decomposition and SEI formation.

**Characterization after lithium plating**. During the initial stage of galvanostatic cathodic polarization of the electrode at 45 °C, the electrolyte gets reduced and the SEI layer forms at the surface

of graphene. This process is then followed by Li plating and possible subsequent SEI reformation upon contact with Li. The $100 \times 100\ \mu$m standard optical image of the G/Li/SEI/SPE composite interface (Fig. 3a) indicates inhomogeneous Li plating. Li deposits vary greatly in shape and size, including Li mounds on the order of ~10 $\mu$m in diameter and interconnected tendrils with widths on the order of ~1 $\mu$m. The high-resolution AFM image (Fig. 3a top right) shows a vastly changed surface morphology that is likely now dominated by a mixture of Li deposits and the SEI layer. The WL image (Fig. 3a bottom right), a measure of local IR reflection, shows significantly more variability and

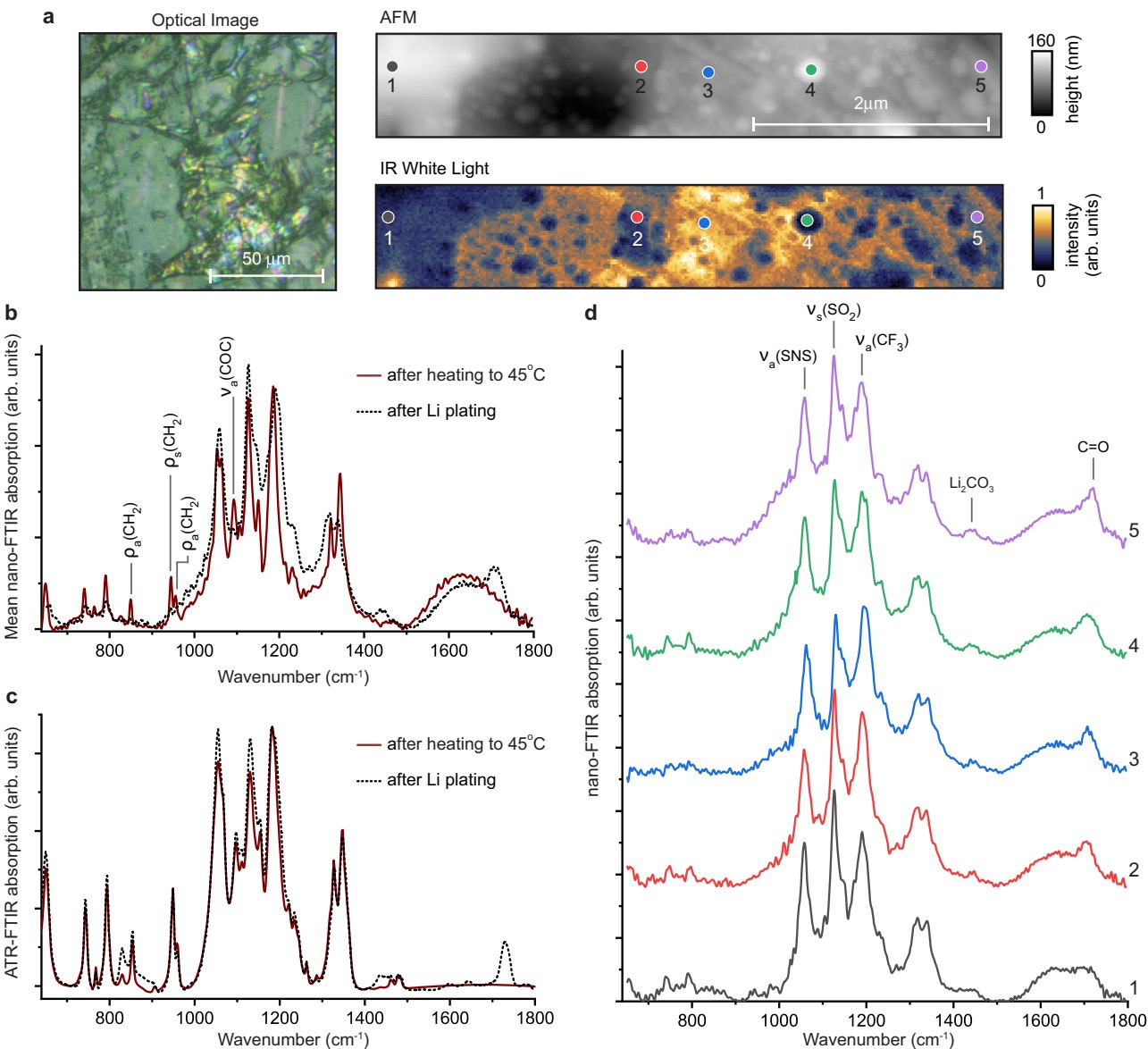

**Fig. 3 Characterization of the G/Li/SEI/SPE interface after Li plating. a** Optical, infrared (IR) white light, and atomic force microscopy (AFM) images after Li plating. **b** Averaged nano-FTIR and **c** ATR-FTIR spectra of the interface before Li plating (the after heating to 45 °C state) and after Li plating. **d** Local nano-FTIR spectra after Li plating (collection locations marked in AFM and WL images).

contrast than the pristine G/SPE interface at room temperature or after heating to 45 °C (Fig. 2a, b). This irregularity arises from inconsistency in interfacial chemical composition and, primarily, differing amounts of locally plated Li: the more Li, the greater the IR reflection (because IR reflection scales with electronic conductivity). This is confirmed with spatially-resolved nano-FTIR reflection spectra (see Methods and Supplementary Fig. 5) and indicates that Li was plated non-uniformly at the surface of G electrode across different length scales.

Comparisons of IR spectra obtained before and after plating reveal chemical changes that occurred at the G/SPE interface during the electrochemical process. Both nano-FTIR (Fig. 3b, d) and ATR-FTIR (Fig. 3c) show the emergence of new IR absorption bands, including features at 1420–1470 cm$^{-1}$ that correspond to -C-O $\nu_{sym}$ and $\nu_{asym}$ vibration modes in $Li_2CO_3$ (Supplementary Fig. 7) and a band at 1715 cm$^{-1}$ that corresponds to $\nu_{sym}$ of an organic carbonyl group (-C=O). The differences between the nano-FTIR and ATR-FTIR spectra after plating are particularly enlightening because they reveal the

changes that occur only at the interfacial region. Notably, new features in the 1210–1310 cm$^{-1}$ region, which are likely attributed to symmetric and asymmetric -C-O vibration modes in an organic compound, appear only in the nano-FTIR spectrum. Moreover, the intensity of -$CH_2$ rocking bands at 850, 948, and 963 cm$^{-1}$ as well as C-O-C stretching modes at 1059, 1098, and 1148 cm$^{-1}$ of crystalline PEO are significantly reduced and broadened only in the nano-FTIR spectrum, indicating the formation of a mixture of organic and inorganic PEO decomposition products at the surface of the electrode. Additionally, these results suggest that the PEO in the bulk electrolyte remains in its original semi-crystalline state and that only a small fraction of the electrolyte adjacent to the G/Li electrode surface has undergone partial decomposition[37] and/or phase transformation from a crystalline to amorphous state[35,38].

An additional difference between the nano-FTIR (Fig. 3b, d) and ATR-FTIR (Fig. 3c) absorption spectra is the observed broad maximum at ca. 1500–1800 cm$^{-1}$. This broad spectral feature, only detected with nano-FTIR, likely arises from optically excited

electronic transitions between graphene and the SPE due to coupling of their electronic band structures. While there is rich physics related to the electronic structures of doped graphene/s and graphene heterostructures[39], and SPEs[40], elucidating the details of the complex interactions and near-surface electronic band structure is beyond the scope of the present work. However, this effect is a direct manifestation of the enhanced sensitivity and surface selectivity of the nano-FTIR probe with regard to the standard ATR-FTIR probing depth and mechanism.

Interestingly, the peak position and intensity of carbonyl and carboxyl bands in Li carbonate and organic decomposition compounds vary strongly with location, as highlighted in Fig. 3d, suggesting nonuniform distribution of the electrolyte decomposition products. Moreover, the three primary bands of the TFSI⁻ anion also display noticeable variability in peak intensity and position. While this may also suggest further chemical heterogeneity, influences from heterogeneous nanoscale Li plating and associated screening of SPE absorption bands is also playing a role (see Methods and Supplementary Fig. 5g,h). In any case, the nonuniform distribution of the electrolyte decomposition products, amounts of Li plating, and G/Li electrode surface (electro) chemical reactivity in all likelihood originate from the intrinsic chemical and molecular structure inhomogeneities observed in the interfacial $PEO_{10}$:LiTFSI electrolyte. Because the thin (nanoscale) metallic Li deposits act to some extent as an optical screen of the near-fields (see Methods and Supplementary Fig. 5), nano-FTIR probing and detection of trace interfacial chemical species may be inhibited unless Li is stripped away: G/Li/SEI/SPE to G/SEI/SPE.

**Characterization after lithium stripping.** The optical image (Fig. 4a) collected after galvanostatic stripping of plated Li shows no visible trace of Li deposits and fewer variations in interfacial roughness. Similarly, the high-resolution AFM image and the corresponding WL picture collected along the surface of the G/SPE heterostructure clearly distinguish large ca. 1–2 μm, fairly uniform and densely packed clusters. Nano-FTIR spectra collected across four neighboring clusters (locations marked in Fig. 4a) can be divided into two distinct groups: "I" - plots 1–4 and 9–12 and "II" - plots 5–8, which show their own common spectral characteristics (Fig. 4d). Averages of each group, color-matched appropriately, are plotted alongside the mean nano-FTIR absorption spectrum of the G/SPE interface after heating to 45 °C, for reference, in Fig. 4b. Interestingly, those spectra from regions I&II and their corresponding locations are inconsistent with the morphology and topology of the interface i.e., flat areas vs. grain boundaries or elevated areas vs. depressions.

Clearly, none of the nano-FTIR spectra resemble the original spectrum of the electrolyte before or after heating to 45 °C and they consistently show new features that belong to the electrolyte decomposition products or the SEI layer. In both "groups" the intensity of C-O-C stretching at ca. 1100 $cm^{-1}$ is greatly reduced as compared to the G/SPE case after heating due to decomposition of PEO. Also, the absence of $CH_2$ symmetric rocking modes at 850, 948, and 963 $cm^{-1}$ of PEO (Fig. 4b,d and Supplementary Fig. 8) implies that it remains amorphous within the SEI whereas the bulk SPE electrolyte remains semi-crystalline (ATR-FTIR, Fig. 4c). An additional observation is the reduction in the maximum of the broad feature which corresponds to electronic transitions between graphene and the SPE at ca. 1500–1800 $cm^{-1}$. This reduction in absorption is likely due to a disruption in G-SPE coupling by a combination of an insulating barrier comprised of the SEI and decomposition products, and a less robust mechanical contact post stripping. As an aside, this later point may indicate these devices are not ideal for long-term

cycling, perhaps due to a lack of stack pressure and mechanical robustness provided from a bulk electrode, as found in conventional SSB devices.

The SEI layer in Region I locations appears to be enriched with TFSI⁻ anions likely arising from anion migration toward the interface during Li stripping. However, the significant change in the relative intensities of the S–N–S asymmetric stretching, C–$SO_2$–N bonding and $CF_3$ asymmetric stretching bands may also indicate severe decomposition of TFSI- and formation of chemically close derivatives with similar functional groups e.g., $LiSO_2CF_3$, $Li_2NSO_2CF_3$, $Li_2S_2O_4$, and $Li_2SO_3$. This TFSI⁻ behavior and reduction mechanism is consistent with previously reported computational and experimental results[41]. In addition, other decomposition products of $Li_2CO_3$, $LiO_2$, and intermediate could be observed. Remarkably, peaks at 1266, 1302, and 1316 $cm^{-1}$ may indicate the presence of Li hydrides, $(LiH)_{x=1,4}$[42], which can form upon reduction of PEO to hydrogen and reaction with metallic Li[37]. Formation of hydrides have also been observed in other Li metal battery systems[43,44] and are known to encumber functionality through the irreversible growth of hydride dendrites. The new notable peak at 1225 $cm^{-1}$ is likely attributed to symmetric and asymmetric -C-O vibration modes in organic compounds.

The nano-FTIR spectra from Region II locations show a visibly different intensity pattern of TFSI⁻-related peaks, which indicate significantly altered anion decomposition reaction mechanisms and rates. Their lower relative intensity with regard to PEO bands may indicate TFSI⁻ depletion, suggesting areas with no or minimal Li dissolution and resulting anion accumulation compared to Region I. Overall, the nano-FTIR absorption spectrum of Region II is somewhat similar to the pristine G/SPE interface after heating, albeit with reduced amounts C-O-C and $SO_2$ stretching modes due to electrolyte decomposition to $Li_2SO_3$[45], LiF[45,46], $Li_2O$[47], $Li_2O_2$[48], and $Li_2S$[49] at 1360, 837, 800–900, 828 and 809, and 645 $cm^{-1}$ respectively. Remarkably, there is no trace of the irreversible formation of Li hydrides, which are attributed to the SEI layer on Li metal. Therefore, we can hypothesize that the electrode surface locations that correspond to Region II spectra consist of primary graphene with small amount or no Li deposits after plating. A Venn diagram (Fig. 4e) depicts some of these differences (wings of the diagram), as well as similarities (central overlapping region) between the two SEI regions.

The observed variations in the SEI layer's local chemical composition are in concert with the previous observation of nonuniform Li plating on the surface of graphene. That said, considering these heterogeneities within the SEI are tied to nanoscale heterogeneities in the physicochemical properties of the SEI layer, such as local passivating mechanism and resulting film ionic conductivity, those inhomogeneities will likely escalate during long term charge–discharge cycling. Based on the experimental observations and analyses outlined above, we put forward the following electrode surface passivation scenario. Intrinsic nanoscale heterogeneities within the interfacial region of the pristine SPE, specifically those of polymer molecular structure (chain orientation) and especially chemical composition (polymer/salt ratio), directly influence local Li⁺ transport properties and give rise to inhomogeneous SEI formation on a congruent spatial scale (as illustrated in Fig. 1c). These trigger, and continuously aggravate, nonuniform current distribution through the SEI and nonuniform Li platting at the electrode/SEI interface in the following cycles. Moreover, the heterogeneous passive film on the G/Li electrode suffers from the mechanical stress and formation of intergranular crevices or cracks, which result from the lattice mismatch between the electrode surface and the SEI

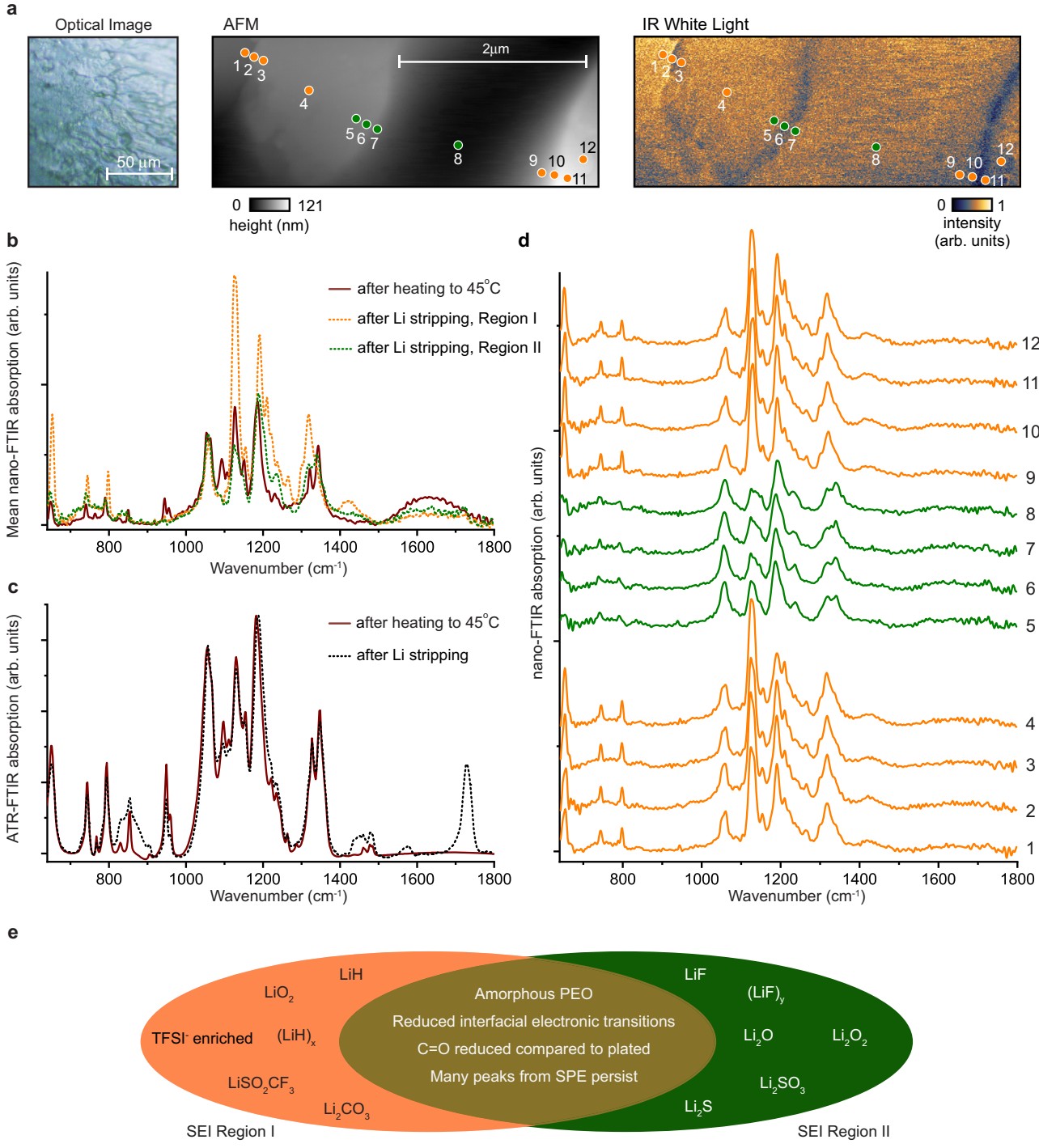

**Fig. 4 Characterization of the G/SEI/SPE interface after Li stripping. a** Optical, atomic force microscopy (AFM), and infrared (IR) white light images of the G/SEI/SPE interface. **b** Average nano-FTIR spectra of the interface after heating to 45 °C and after Li stripping from the two SEI regions indicated by the colored locations in **a. c** ATR-FTIR spectra of the interface after heating to 45 °C and after Li stripping. **d** Individual nano-FTIR spectra of the G/SEI/SPE interface after Li stripping. Spectra are numbered and color-coded according to the locations in **a. e** Venn diagram depicting key similarities and differences between the two predominate SEI regions.

layer chemical building blocks. Ultimately, all these local inhomogeneities, where the film may become less electronically insulating or more ionically conducting, provide pathways for further electrolyte decomposition at the interface between the passivation layer and the electrolyte. This process appears to be at the origin of the commonly observed degradation mechanisms in rechargeable Li metal batteries, which involve continuous electrolyte consumption and Li dendrites formation.

This study also demonstrates that even an atomically flat surface of graphene in contact with an SPE is still prone to develop a heterogeneous surface film composition and structure, which then results in nonuniform Li-plating currents and distributions at the nano- and micro-scale. In fact, the local composition and structure of the electrolyte in the region adjacent to the surface of the electrode, at a molecular level, as well as its micro-scale morphology and topography, need to be precisely

designed to exhibit perfectly uniform physicochemical characteristics. Only then can the observed degradation of Li/SEI/electrolyte interfaces be prevented or inhibited to ensure long-term operation of Li SSB cells. Finally, the unique in situ experimental approach described in this study may be helpful for obtaining the necessary level of control required for the design and realization of such stable electrochemical interfaces needed in future rechargeable batteries.

## Methods

**Battery cell composition, assembly, and testing protocols.** The PEO/LiTFSI based solid-state cell was prepared with an EO/Li ratio of 10:1. The stoichiometric amount of PEO (Sigma Aldrich, average Mv 100,000) and LiTFSI (Sigma Aldrich, 99.95%) was co-dissolved in acetonitrile (ACN) solvent and cast onto Li foil within an argon glovebox. After ACN fully evaporated (confirmed by ATR-FTIR), monolayer graphene (MSE Supplies) was transferred onto the SPE surface, setting up a full cell.

In the following, we quantitatively estimate the amount of Li that is galvanostatically plated to the graphene surface. Theoretically, Li electrodeposition can occur when the anode (graphene) potential becomes less than or equal to the equilibrium potential $E_{Li^+/Li^0}$ of the Li redox reaction: $Li^+ + e^- \rightarrow Li^0$. If we assume a perfectly homogeneous deposition on a flat, idealized current collector with planar structure, a plating thickness of $1\mu m$ would result (calculated based on a constant current of $-0.025$ mA cm$^{-2}$ for 10 h).

**Infrared nanospectroscopy setup, data acquisition, and processing.** In general, nano-FTIR is realized by illuminating, with IR light, an oscillating metallic AFM probe tip and sample placed in one arm of an asymmetric Michelson interferometer. The back-scattered light is combined with a reference beam reflected off a moving mirror in the other arm of the interferometer. A small amount of the backscattered light (collected in the far-field) is attributable to scattering events resulting from near-(electric) field-induced excitations confined to nanoscopic regions below the probe tip end. The backscattered light from localized nanoscopic regions is isolated by a combination of lock-in demodulation (at the second or higher harmonics of the tip-tapping frequency) and normalization (to a spectrally flat material such as silicon or gold) and in this way nondestructive, truly nanoscale IR spectroscopy (or imaging) can be conducted. All nano-FTIR absorption (reflection) spectra, unless otherwise specified, are the second harmonic of the imaginary part (amplitude) of the complex spectrum, referenced to silicon, and are averages of 12 spectra all acquired at a spectral resolution of 6.25 cm$^{-1}$. More details on IR nanospectroscopy (or imaging) can be found in the literature[50].

Near-field nano-FTIR measurements were collected with a commercial Neaspec system equipped with the broadband "nano-FTIR" laser source. The Neaspec system was enclosed within a custom-built glovebox filled and purged with nitrogen. This procedure reduced O$_2$ concentrations to a base level of ~10 ppm. Supplementary Fig. 1 shows spectra of a Si reference sample under ambient conditions and in the N$_2$-filled environmental chamber. Near-field measurements were conducted using commercial "nano-FTIR probes" from Neaspec. Presented spectra over the range 640–1800 cm$^{-1}$ in the main text are stitched together from two spectra utilizing laser ranges "A" (~640–1400 cm$^{-1}$) and "C" (~1100–1800 cm$^{-1}$) of the broadband laser. Modest phase corrections were employed. All near-field data presented in this work were collected under the aforementioned conditions except for three cases. The exceptions were for near-field spectra collected at Lawrence Berkeley National Laboratory's Advanced Light Source, Beamline 2.4. These spectra are explicitly identified in Supplementary Figs. 2 and 6.

**Analysis of infrared nanospectroscopy data in Fig. 2e.** For each nano-FTIR spectra, the six absorption bands were fit (note the PEO COC antisymmetric stretching vibration analyzed was the mode around 1093 cm$^{-1}$) and quantified for each spectrum. The mean spectra for room temperature and after heating to 45 °C were calculated. Then, for each location (notated as a "location index" along the abscissas in Fig. 2e), the percent differences between the six local peak values and corresponding mean peak values were obtained. Subsequent analysis of these data sets reveals that the standard deviations in peak percent differences decrease by 34% post heating. This finding is visualized by comparing the sizes of the overlaid colored bars which highlight regions of plus/minus two standard deviations in the respective data sets. These results demonstrate that while the chemistry of interfacial SPE becomes more homogenous post heating, inhomogeneity at a spatial scale on the order of about 100 nm persists.

**Analysis of infrared nanospectroscopy data across SiO$_2$.** Nano-FTIR spectra are sensitive to the locally averaged molecular orientations within nanoscopic volumes below the probe tip end. Therefore, in order to claim that the nano-FTIR variations measured across the G/SPE interface are significant (e.g., those in Fig. 2 and Supplementary Fig. 5), and not just arising from systemic variations, a control experiment needed to be done. In particular, one which would quantify the minimum amount of systemic nano-FTIR spectra variation which is to be expected

for all spatially dependent measurements. Then, the sample of interest's spatially dependent nano-FTIR variation only becomes meaningful if it clearly exceeds the minimum expected variation.

The ideal control sample would be crystalline and should possess limited to no IR absorption/reflection variation theoretically. The sample we chose that fit this bill was thermally grown SiO$_2$ and the data is displayed in Supplementary Fig. 6 a–f. The results indicate that the expected minimum variation (for spectra collected with separation distances on the order of 100 nm) is less than 2% (3%) for the average local percent difference with respect to mean nano-FTIR absorption (reflection) (Supplementary Fig. 6g, h). In that the similarly measured variations along the G/SPE are roughly an order of magnitude larger than the minimum values expected (Supplementary Fig. 6g,h), we can confidently assert that these variations do not arise from systemic variability implicit to nano-FTIR data collection, but rather from significant heterogeneities in local physicochemical properties as described in the main text.

**Nanoscale heterogeneity in interfacial salt concentration.** Variations in the nano-FTIR absorption spectra collected at the pristine G/SPE interface, and after heating to 45 °C, are helpful for learning how chemistry, crystallinity, and molecular conformations may vary at the surface of the SPE which contacts the electrode. Of particular importance to subsequent electrochemical processes is what may be learned about the relative concentration between the effective solvent and salt (PEO$_n$:LiTFSI) because relative salt concentration considerably influences Li$^+$ conductivity in SPEs[5,36]. For this, we consider the asymmetric stretching mode of SO$_2$ in the TFSI anion. This mode splits into a doublet of "in-phase" and "out-of-phase" bands[33], and their ratio changes in a characteristic way with salt concentration[35] (Supplementary Fig. 4). These two spectral features are clearly resolved in the nano-FTIR absorption spectra (main text Fig. 2 c,d) around 1322 and 1343 cm$^{-1}$, and their ratio (Supplementary Fig. 4) is highly variable, implying that nanoscale heterogeneity of salt concentration—and correlated ionic conductivity—is intrinsic to the surfaces of these SPEs which contact electrodes. While heating reduces the variability, it does not remove it. Finally, as shown in Supplementary Fig. 4, similar ratios are taken from our ATR-FTIR measurements and match what is expected from the literature for a bulk concentration of PEO$_{10}$:LiTFSI.

**Nanoscale heterogeneity in interfacial Li plating.** Before Li plating, near-field IR WL imaging and nano-FTIR reflection measurements were collected (Supplementary Fig. 5a,b). Reflection spectra, in wavenumber regions unobstructed by molecular vibrations (e.g., 668–725 cm$^{-1}$), indicate local electronic properties of the interfacial SPE are relatively similar, in agreement with the results of the WL image which displays a uniform response. However, after plating, near-field IR WL imaging and nano-FTIR reflection data (Supplementary Fig. 5c,d) have significantly more variability, suggesting the electronic conductivity of the plated interface is heterogenous at the nanoscale. Furthermore, there is a linear correlation between average nano-FTIR reflection (in wavenumber regions unobstructed by molecular vibrations) and local WL image intensity (Supplementary Fig. 5e). This confirms that WL image contrast in Supplementary Fig. 5c is directly visualizing variability in reflection, which, in this system, arises from heterogeneity in Li plating and associated local electronic conductivity: brighter spots on the WL images correspond to more Li locally deposited at the G/Li/SPE interface. Supplementary Fig. 5f helps to quantitatively show that reflection variability significantly increases between the heated G/SPE interface and the plated G/Li/SPE interface, as should be expected from heterogenous Li plating.

**Screening of SPE absorption bands by plated Li.** We now outline an additional physical phenomenon observed that is consistent with nanoscale heterogenous Li plating at the G/Li/SPE interface: screening of SPE absorption bands by overlaid G/Li. Due to plating of Li on the graphene current collector, characteristic peaks of the buried SPE at lower wavenumbers (<1200 cm$^{-1}$) seem to be suppressed based on the amount of local Li plated. This is most easily seen in Supplementary Fig. 3g. Consider, for example, the stretching mode of CS at about 794 cm$^{-1}$. Here, it is easily seen that less (more) absorption is measured at spatial locations with more (less) measured reflection: the blue and purple (gray and green) spectra are collected at bright (dark) locations on the WL image (Supplementary Fig. 5c). The suppression also seems to be frequency dependent: the lower the wavenumber, the greater the suppression. This is verified through a quantitative normalization process described in the following, and whose results are plotted in Supplementary Fig. 5h.

In Supplementary Fig. 5h, nano-FTIR reflection (again, averaged over the wavenumber range 668–725 cm$^{-1}$) is used for the horizontal axis and serves as a proxy for the relative amount of local Li plated. Presented in a scatterplot are absorption peak values normalized to corresponding peak values measured at the location with least reflection/plating. Linear fits are overlaid on data for each vibrational mode. The slopes are negative (i.e., more suppression with thicker Li plating) and decrease in absolute value toward a slope of zero as the center frequency of the mode increases in wavenumber (increasing frequency reduces the screening effect). The slope of the normalized $\nu_a$CF$_3$ mode (peak center at 1186 cm$^{-1}$) is close to zero, indicating that absorption bands at or above this frequency are unaffected by this apparent screening process. Of course, these conclusions are only valid for "small"

plating thicknesses, and in the limit of "large" amounts of Li plating, all peaks should be completely suppressed as the near-fields would not penetrate thick metallic films.

## Data availability

The data that support the findings of this study will be made available by the corresponding authors upon reasonable academic request within the limitations of informed consent.

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

## Acknowledgements

R.K. and X.H. kindly acknowledge the financial support of Assistant Secretary for Energy Efficiency and Renewable Energy, Vehicle Technologies Office, under the Advanced Battery Materials Research (BMR) Program, of the U.S. Department of Energy under Contract No. DE-AC02-05CH11231. Funding to support this work was provided to J.M.L., H.A.B., and R.K. by the Energy & Biosciences Institute through the EBI-Shell program. Additionally, this research used resources of the Advanced Light Source, a U.S. DOE Office of Science User Facility under contract no. DE-AC02-05CH11231. In particular, Beamline 2.4 was utilized.

## Author contributions

R.K. conceived the project concept. J.M.L, H.A.B., and R.K. supervised the project. X.H. designed and operated the in situ electrochemical cell. X.H., J.M.L. carried out the experiments and initial data analysis. J.M.L. performed the near-field data analyses. All authors contributed to the interpretation, conclusions, and preparation of the manuscript. X.H. and J.M.L. are listed in alphabetical order as equally contributing authors.

## Competing interests

The authors declare no competing interests.
