## [Peer Review File · Nature Communications]

REVIEWER COMMENTS

Reviewer #1 (Remarks to the Author):

The authors use near-field infrared nanospectroscopy to reveal the intrinsic molecular, structural and chemical heterogeneities of graphene/Li/solid polymer electrolyte interface. The design is smart where the single-layer sheet operates as both an infrared transparent window and a current collector for Li plating and stripping. Therefore, it is possible to unveil and monitor evolution of the interface at different stages during the charge-discharge process. I recommend a major revision of this manuscript. The followed concerns should be further addressed:

1. The measurements were performed at the end of each stage after the room temperature was reached (line 80-82, page 2). The authors also didn't mention whether the applied potential is still on during measurements. It is possible that the interfacial state changes when the temperature decreases and the applied potential is off. In this case, it is not so appropriate to call it a "in situ" measurement. If the authors is able to obtain the spectroscopy during the whole process shown in Fig. 1b, it will really provide some new insight about the interface.

2. Using nano-FTIR, the authors obtained different spectral features at different location after Li plating and stripping. I'm wondering why the authors choose to measure at some selected points other than give a nano-IR mapping of the interface which can show a comprehensive distribution of the chemical heterogeneities. In addition, the heterogeneities of both bulk materials and interface and the resulting inhomogeneous electrochemical behavior on the interface is well known, how can we use these advanced technique to provide some new understanding?

3. The authors mentioned the adhesion between the polymer surface and graphene sheet allows graphene to reproduce the morphology of the underlying polymer (line 94-95, page 4). From the AFM image shown in Fig. 2a and 2b, the height difference is up to 447 nm (before heating) and 140 nm (after heating). How the authors assure that there are no bubbles between two layers and no wrinkles or folds of graphene. All these heterogeneities will also significantly influence the Li plating and stripping process.

4. Page 6, line 138-139, the authors describe "The high-resolution AFM image (Fig. 3a top 138 right) shows the surface morphology that is dominated by a mixture of lithium deposits and the SEI layer." I can't get the point that why the AFM image can tell the chemical composition of the surface.

In addition, there are several peaks analyzed in different conditions and the IR spectrum shows more peaks. It will be much easier to read if the authors can label the wavenumber on the spectra, otherwise it is hard for readers to find the peaks.

Reviewer #2 (Remarks to the Author):

The article reports on a unique approach to combine ATR-FTIR and Nano-FTIR techniques for characterization over the electrode electrolyte interface/interphase for solid electrolytes in Li-Ion batteries. The results and methodology are significant to the field and related battery work specially for SEI characterization. The experimental approach was nicely conceived, and the combination of multi-scale techniques adds a great value for a more refined discussion. The paper is also well written and provides sufficient supporting evidence to support the main claims.

A few observations and question however must be made to fully attest the expected high quality of Nature Communication.

Comment 1: Authors correctly say that “Because of the large amount of charge and mass transferred between electrodes in a battery over its lifetime, the electrode materials and interfaces tend to evolve with time, and in general, degrade in performance, resulting in eventual battery failure” However the tool presented by the authors, although very interesting, still remains unable to access the interphase in a more realistic cell and thus to access degradation in performance/failure diagnostics. We realise that the cell design was probably the best possible option to allow for the technique, but perhaps a few words regarding the limitations or regarding to expected differences to an interface found on a Solid-state device would better inform the readers.

Comment 2: Authors list a variety of techniques to probe interfaces (line 48). Could the authors provide 1 or 2 lines regarding their limitations, or in which sense the techniques presented in the paper adds over those listed ones?

Comment 3: Fig1 shows the cell construction and how a single graphene layer is used. In this regard, folding was an issue to build the electrode? Also, does the graphene layer might act as a driving agent for smoothing out the surface when heating it?

Comment 4: Authors mention that relative intensities of the nano-FTIR spectra varies with location and attribute them to inhomogeneities in the PEO-TFSI-, however Nano-FTIR tends to present more variation than ATR or other IR techniques due to the nano-domains and molecular orientation, therefore it would be interesting to know if duplicate or triplicate spectra's have been taken at each point, or if this was not an issue, please provide some explanation to why it would not be the case in the analysis.

Comment 5: Authors highlight the importance to analyze the differences seen between ATR and SNOM (line 152), specially the -CH₂ rocking bands and -C-O vibrations, these spotted differences should be also highlighted in Fig 3b and 3c for easier visualization.

Comment 6: in line 236 authors claim that the PEO/TFSI- heterogeneities are tied to nanoscale, but I guess this cannot be said by the results on Fig2 as the spotted differences were identified in a larger length-scale.

Comment 7: In line 242 authors say that “inhomogeneities will likely escalate during long term charge-discharge cycling” Although it is highly likely to be true, is it possible to use the proposed cell design on Fig1 to perform a few cycles? Or other instabilities might prevent the experiment?

We would like to thank Reviewers for their constructive criticism, comments and suggestions. Please find below our point-by-point responses to the Reviewers' comments.

Reviewer #1

The authors use near-field infrared nanospectroscopy to reveal the intrinsic molecular, structural and chemical heterogeneities of graphene/Li/solid polymer electrolyte interface. The design is smart where the single-layer sheet operates as both an infrared transparent window and a current collector for Li plating and stripping. Therefore, it is possible to unveil and monitor evolution of the interface at different stages during the charge-discharge process. I recommend a major revision of this manuscript. The followed concerns should be further addressed:

We appreciate the reviewer's kind words acknowledging the uniqueness and utility of our experimental setup.

1. The measurements were performed at the end of each stage after the room temperature was reached (line 80-82, page 2). The authors also didn't mention whether the applied potential is still on during measurements. It is possible that the interfacial state changes when the temperature decreases and the applied potential is off. In this case, it is not so appropriate to call it a "in situ" measurement. If the authors is able to obtain the spectroscopy during the whole process shown in Fig. 1b, it will really provide some new insight about the interface.

The Reviewer is absolutely right when saying that the measurements were performed at room temperature with the applied potential off. In fact, the text already clearly stated that the measurements were performed at room temperature, but for the sake of clarity we have added additional language to explicitly indicate that the measurements were performed with the applied potential off.

The reviewer is correct that additional changes to the interface as a result of thermalization or drift in the potential difference cannot be ruled out. For example, the solid-state electrolyte may partly recrystallize with reduced temperature and reactions may continue without an applied potential. However, the rate of such reactions at RT would be substantially lower than at 45°C due to slower kinetics and lower diffusion rates in the solid electrolyte. Therefore, we expect that the corresponding changes at the interface are not only negligible but also consistent with the processes that occurred at 45°C. For the same reason, we do not expect any significant drift of the electrode potential of the bare graphene electrode before Li plating and after Li stripping. The Li-plated graphene electrode should maintain its potential constant by definition. In fact, we note that replicate measurements at the same location showed no evidence of spectral changes with time, so if there were any changes to the sample, they occurred before the nano-FTIR measurements were performed.

We agree with the reviewer that "in operando" nano-FTIR measurements conducted along the whole electrochemical process would be extremely insightful. However, for the current experimental parameters, in operando measurements are just beyond the state-of-the-art, primarily due to the fact that the elevated heat applied to the sample causes AFM cantilever instabilities (largely from convection processes), resulting in prohibitively high noise. We believe though that our measurements still qualify as "in situ" measurements because the cell and

interfaces are still intact and no factors or interventions, which could affect its original function and mechanism of operation took place. This word choice recognizes the literal definition of “in situ”: in the natural or original position or place, which refers to the location of measurements and not the operating conditions.

2. Using nano-FTIR, the authors obtained different spectral features at different location after Li plating and stripping. I'm wondering why the authors choose to measure at some selected points other than give a nano-IR mapping of the interface which can show a comprehensive distribution of the chemical heterogeneities. In addition, the heterogeneities of both bulk materials and interface and the resulting inhomogeneous electrochemical behavior on the interface is well known, how can we use these advanced technique to provide some new understanding?

We agree with the reviewer that hyperspectral imaging with nano-FTIR over a large area could show a comprehensive distribution of the chemical heterogeneities. However, the time to achieve high quality broadband spectra with high spatial resolution was time prohibitive for the signal to noise ratio of our current system. Collecting a square map over similar surface areas (say 6 by 6 microns) at 100 nm (10 nm) resolution would have taken 210 hours (21,000 hours) to achieve the same quality spectra. As another practical limitation, our laser source has a narrow spectral bandwidth, so each broadband spectrum presented in the work is actually spliced by two independent spectra collected at same point of interest: one over the range $\sim 640 - 1400 \text{ cm}^{-1}$ and another over the range $\sim 1100 - 1800 \text{ cm}^{-1}$ (as described in the Methods section under the subsection “IR nanospectroscopy setup, data acquisition and processing overview”). The two spectra need to be taken within a short timeframe so as to minimize in-plane piezo drift and two spectra should be collected over the same spot to ensure the matchable spectra in the overlapping range. Current mapping modes supported by the software of our near-field tool do not allow mapping in which two spectra (over different wavenumber ranges) are collected at each location. For these reasons, we chose instead to prioritize collecting high quality broadband spectra at single nanoscale points combined with complementary white-light imaging. The combined measurement modalities allow more time-efficient probing of the intact interface and yields similar information that would be obtained with a full hyperspectral image. We note, however, that IR laser sources are rapidly improving in bandwidth, intensity, and stability, and we anticipate that hyperspectral nano-imaging will be possible in the future using the platform we have demonstrated.

While we agree with the reviewer that the electrochemical behaviors will be affected by heterogeneities of electrolyte have been known for a while, we believe this unique work provides a pathway to directly verify and interrogate such phenomena (i) with nanoscale resolution, (ii) in a non-destructive fashion (due to the low-energy of the IR photons), and (iii) within the interface's buried and intact native environment. We are unaware of any other study and/or methodology that has been able to study electrochemical interfaces and interphases in such a way that simultaneously achieves all three of these important characterization traits. Moving forward, we envision, in the least, opportunities for additional basic science understanding to expand by exploring other systems, operando work, multiple cycles, and characterization efforts needed for the R&D of engineered interfaces.

3. The authors mentioned the adhesion between the polymer surface and graphene sheet allows graphene to reproduce the morphology of the underlying polymer (line 94-95, page 4). From the AFM image shown

in Fig. 2a and 2b, the height difference is up to 447 nm (before heating) and 140 nm (after heating). How the authors assure that there are no bubbles between two layers and no wrinkles or folds of graphene. All these heterogeneities will also significantly influence the Li plating and stripping process.

The reviewer makes a good point that graphene abnormalities such as wrinkles and bubbles would significantly influence the local electrochemistry, especially in the case of bubbles, which would effectively disconnect the local current collector (graphene) from the electrolyte. We shared this concern as well, and we addressed the issue during the sample preparation process and during the measurement, as reasonably as possible. We checked the optical image first and then conducted AFM over each potential region prior to commencing with nano-FTIR measurements, effectively pre-screening out any region which displayed structures which may have indicated the graphene was locally compromised in one of the two aforementioned ways. We conduct the nano-FTIR measurements only after a region cleared this test. All that said, the solid-state electrochemical interface is hard to reach 100% perfectly homogenous physical connection, but this is true for real world devices as well. Indeed, the flexibility of an atomically thin layer of graphene is one of the best choices to optimize such interfacial connectivity.

4. Page 6, line 138-139, the authors describe “The high-resolution AFM image (Fig. 3a top 138 right) shows the surface morphology that is dominated by a mixture of lithium deposits and the SEI layer.” I can’t get the point that why the AFM image can tell the chemical composition of the surface.

The reviewer is absolutely correct that chemical composition cannot be directly gleaned from AFM, and we certainly don’t want a reader to think this is what we are saying. We have therefore adjusted the language in the main text to make our point clearer. “The high-resolution AFM image (Fig. 3a top right) shows a vastly changed surface morphology that is likely now dominated by a mixture of lithium deposits and the SEI layer.” The intention of the sentence was to stress that the measured morphology, post galvanostatic plating, should not be dominated by intrinsic structural heterogeneities of the solid polymer electrolyte, but by a combination of plated Li and the SEI. This assertion isn’t controversial, but is supported in a number of ways in the analyses that follow the statement.

5. In addition, there are several peaks analyzed in different conditions and the IR spectrum shows more peaks. It will be much easier to read if the authors can label the wavenumber on the spectra, otherwise it is hard for readers to find the peaks.

We thank the reviewer for pointing this out. As also recommended by reviewer 2 in comment 5, we have included additional labels on Figure 3. We hope this helps aid the reader. Moreover, we have also made an additional Supplementary Figure 8, which should also help readers seeking greater detail to more quickly identify spectral features.

Reviewer #2

The article reports on a unique approach to combine ATR-FTIR and Nano-FTIR techniques for characterization over the electrode electrolyte interface/interphase for solid electrolytes in Li-Ion batteries. The results and methodology are significant to the field and related battery work specially for SEI characterization. The experimental approach was nicely conceived, and the combination of multi-scale techniques adds a great value for a more refined discussion. The paper is also well written and provides sufficient supporting evidence to support the main claims.

A few observations and question however must be made to fully attest the expected high quality of Nature Communication.

We appreciate the praise related to the quality and relevancy of our work, as well as the helpful suggestions.

Comment 1: Authors correctly say that “Because of the large amount of charge and mass transferred between electrodes in a battery over its lifetime, the electrode materials and interfaces tend to evolve with time, and in general, degrade in performance, resulting in eventual battery failure” However the tool presented by the authors, although very interesting, still remains unable to access the interphase in a more realistic cell and thus to access degradation in performance/failure diagnostics. We realise that the cell design was probably the best possible option to allow for the technique, but perhaps a few words regarding the limitations or regarding to expected differences to an interface found on a Solid-state device would better inform the readers.

The reviewer makes a fair point that the device configuration as presented may not be optimized for studying degradation in performance over long-term cycling, and that there are discrepancies between our device and an actual solid-state battery (e.g. our graphene window cannot exert any meaningful stack pressure). We have adjusted the language in the main text to highlight some of the implications of our cell design not exactly mimicking that of a commercial SSB. “As an aside, this later point may indicate these devices are not ideal for long-term cycling, perhaps due to a lack of stack pressure and mechanical robustness provided from a bulk electrode, as found in conventional SSB devices.” That said, the primary aim of this work was not to study interface/interphase degradation over the lifetime of a battery. Rather, our aim was to advance, and apply, a unique methodology able to unveil underlying physicochemical properties of buried and intact interfaces over microscale areas with nanoscale resolution; especially properties critical to explaining the origins of heterogeneous Li-ion kinetics and early-stage SEI formation, and properties of SEI chemistry and structure. Toward this end, we feel we were quite successful. (Please also see our response to Comment 2 as it is relevant)

Comment 2: Authors list a variety of techniques to probe interfaces (line 48). Could the authors provide 1 or 2 lines regarding their limitations, or in which sense the techniques presented in the paper adds over those listed ones?

Thank you for this comment. We do believe the technique presented in this work presents a number of notable advancements over previous work. We have changed the language in the corresponding section to better reflect these advantages. “However, no methodology has been

demonstrated thus far that can characterize interface and SEI nondestructively, within their undisturbed native environment, and with nanoscale resolution. Here, we present an in situ methodology capable of just that.” As described in our response to Reviewer 1’s second comment, our technique provides a pathway to study interfacial phenomena and physicochemical properties (i) with nanoscale resolution, (ii) in a non-destructive fashion (due to the low-energy of the IR photons), and (iii) within the interface’s buried and intact native environment. We are unaware of any other study and/or methodology which has been able to study electrochemical interfaces and interphases in such a way that simultaneously achieves all three of these important characterization traits.

Comment 3: Fig1 shows the cell construction and how a single graphene layer is used. In this regard, folding was an issue to build the electrode? Also, does the graphene layer might act as a driving agent for smoothing out the surface when heating it?

Graphene folding. As described in our response to reviewer 1’s 3rd comment, it is impossible to prevent some level of structural abnormalities from naturally arising when assembling the cell. That said, we avoided performing nano-FTIR across areas likely affected with such abnormalities by pre-screening potential areas of interest with optical image and AFM. Thus, we did not find folding to be a significant issue in this work.

Graphene and smoothing. While we cannot completely preclude graphene from acting as a catalyst of sorts for surface smoothing during heating, we feel the most likely contributor to this observation is the known enhanced fluidity of the polymer at elevated temperatures.

Comment 4: Authors mention that relative intensities of the nano-FTIR spectra varies with location and attribute them to inhomogeneities in the PEO-TFSI- , however Nano-FTIR tends to present more variation than ATR or other IR techniques due to the nano-domains and molecular orientation, therefore it would be interesting to know if duplicate or triplicate spectra’s have been taken at each point, or if this was not an issue, please provide some explanation to why it would not be the case in the analysis.

The reviewer is quite correct that nano-FTIR spectra are sensitive to the locally averaged molecular orientation. It is also true that nano-FTIR spectra often present more variation than ATR-FTIR due to systemic reasons. Therefore, in order to claim that the nano-FTIR variations measured across the G/SPE interface are significant (e.g. those in Fig 2), and not just arising from systemic variations, a control experiment needed to be done. In particular, one which would quantify the minimum amount of systemic nano-FTIR spectra variation that is to be expected for all spatially dependent measurements. Then, the sample of interest’s spatially dependent nano-FTIR variation only becomes meaningful if it clearly exceeds the minimum expected variation.

The ideal control sample would be crystalline and should possess limited to no IR absorption/reflection variation theoretically. We chose thermally grown SiO₂ as the control sample, and this control experiment is mentioned in the main text and the results are displayed in supplementary Fig. 6.

The results indicate that the expected minimum variation (for spectrum collected with separation distances on the order of 100 nm) is less than 2% (3%) for the average local percent difference with respect to mean nano-FTIR absorption (reflection). In that the similarly measured variations along the G/SPE are roughly an order of magnitude larger than the minimum values expected, we

can confidently assert that these variations do not arise from systemic issues, but rather from significant heterogeneities in local physicochemical properties. In light of the reviewer's concern, we have made these points available to the reader by incorporating a majority of this response into a new Methods section titled "Analysis of spatially-resolved nano-FTIR spectra on a chemically homogenous material."

Related to the number of spectra taken per point: as described in our response to reviewer 1's second comment, at each location two spectra were taken with different wavenumber ranges, and stitched together to make a broadband spectra. The two spectra had approximately three hundred wavenumber overlap which matched well. Furthermore, each "single" spectrum is actually an average of 12 spectra collected sequentially, and we now state that clearly in the relevant methods section. So, the answer to the reviewer's question is yes, we took multiple spectra at each location.

Comment 5: Authors highlight the importance to analyze the differences seen between ATR and SNOM (line 152), specially the -CH₂ rocking bands and -C-O vibrations, these spotted differences should be also highlighted in Fig 3b and 3c for easier visualization.

Thank you for this helpful comment, we have incorporated your suggestion into the Figure 3 to aid future readers.

Comment 6: in line 236 authors claim that the PEO/TFSI- heterogeneities are tied to nanoscale, but I guess this cannot be said by the results on Fig2 as the spotted differences were identified in a larger length-scale.

The reviewer is correct that the separation distance between nano-FTIR points are different in each of the figures, however we feel the language is warranted because of the following reasons: (i) all of the separation distances are on the order of 100 nm, and also because (ii) the localized sampling volume is known to be nanoscale.

Moreover, s-SNOM white light images provided in our work provide additional near-field IR data sets. These possess a spatial resolution limited by the radius of curvature of the tip (~10nm). As described briefly in the main text, and in detail in the Methods subsection titled "Nanoscale heterogenous Li plating at the G/Li/SPE interface confirmed with near-field IR WL Imaging, nano-FTIR reflection and absorption, and related analyses.", nanoscale variations in the white light images correspond to variations in nano-FTIR reflection (which correlates with local chemistry).

Comment 7: In line 242 authors say that "inhomogeneities will likely escalate during long term charge-discharge cycling" Although it is highly likely to be true, is it possible to use the proposed cell design on Fig1 to perform a few cycles? Or other instabilities might prevent the experiment?

Running experiments with our proposed cell design over a few additional cycles may be doable, but is not the focus of this work. As described in detail in our response to reviewer 2's first comment, our aim is to demonstrate a unique methodology, which is able to unveil underlying physicochemical properties of buried and intact interfaces over microscale areas with nanoscale

resolution. With the spectra collected from the representative points during the first cycle, we could explain the origins of heterogeneous Li-ion kinetics, morphological and chemical properties of the SEI layer.

REVIEWERS' COMMENTS

Reviewer #1 (Remarks to the Author):

The authors well addressed all the concerns. I agree with the authors that this technique provides a pathway to study electrochemical interfacial phenomena and physicochemical properties (i) with nanoscale resolution, (ii) in a non-destructive fashion (due to the low-energy of the IR photons), and (iii) within the interface's buried and intact native environment. I recommend a publication of this manuscript.

Reviewer #2 (Remarks to the Author):

The authors provided a much-improved version from the original manuscript with a clearer connection between the data, a better introduction stating the benefits the importance and the unique of the work and a few important limitations were also highlighted. I appreciated the effort on providing a detailed description for the SiO₂ standard (in methods and in Fig S6) that for sure will help a much broader audience to better realize the intricacies of the nano-FTIR analysis. The provided responses for all queries were satisfactorily and therefore I recommend in favor of accepting the work as presented in this new version.

Point-by-Point Response to the Reviewers' Comments

REVIEWERS' COMMENTS

Reviewer #1 (Remarks to the Author):

The authors well addressed all the concerns. I agree with the authors that this technique provides a pathway to study electrochemical interfacial phenomena and physicochemical properties (i) with nanoscale resolution, (ii) in a non-destructive fashion (due to the low-energy of the IR photons), and (iii) within the interface's buried and intact native environment. I recommend a publication of this manuscript.

We appreciate your perspective on the uniqueness and importance of our work. Moreover, we sincerely thank you for the time and effort put into reviewing our manuscript. Your insightful inquiries and suggestions ultimately aided in the optimization of our work.

Reviewer #2 (Remarks to the Author):

The authors provided a much-improved version from the original manuscript with a clearer connection between the data, a better introduction stating the benefits the importance and the unique of the work and a few important limitations were also highlighted. I appreciated the effort on providing a detailed description for the SiO₂ standard (in methods and in Fig S6) that for sure will help a much broader audience to better realize the intricacies of the nano-FTIR analysis. The provided responses for all queries were satisfactorily and therefore I recommend in favor of accepting the work as presented in this new version.

We are glad to hear that you feel the latest version of our manuscript is significantly improved on a number of fronts. Moreover, we sincerely thank you for the time and effort put into reviewing our manuscript. Your insightful inquiries and suggestions ultimately aided in the optimization of our work.